# Deep Learning Based Apples Counting for Yield Forecast Using Proposed Flying Robotic System

**DOI:** 10.3390/s23136171

**Published:** 2023-07-05

**Authors:** Şahin Yıldırım, Burak Ulu

**Affiliations:** Department of Mechatronic Engineering, Erciyes University, Kayseri 38039, Turkey

**Keywords:** deep learning, agricultural automation, aerial robotics, object detection, computer-vision

## Abstract

Nowadays, Convolution Neural Network (CNN) based deep learning methods are widely used in detecting and classifying fruits from faults, color and size characteristics. In this study, two different neural network model estimators are employed to detect apples using the Single-Shot Multibox Detection (SSD) Mobilenet and Faster Region-CNN (Faster R-CNN) model architectures, with the custom dataset generated from the red apple species. Each neural network model is trained with created dataset using 4000 apple images. With the trained model, apples are detected and counted autonomously using the developed Flying Robotic System (FRS) in a commercially produced apple orchard. In this way, it is aimed that producers make accurate yield forecasts before commercial agreements. In this paper, SSD-Mobilenet and Faster R-CNN architecture models trained with COCO datasets referenced in many studies, and SSD-Mobilenet and Faster R-CNN models trained with a learning rate ranging from 0.015–0.04 using the custom dataset are compared experimentally in terms of performance. In the experiments implemented, it is observed that the accuracy rates of the proposed models increased to the level of 93%. Consequently, it has been observed that the Faster R-CNN model, which is developed, makes extremely successful determinations by lowering the loss value below 0.1.

## 1. Introduction

Apples have a wide range of uses in the food industry around the world. Worldwide, annual apple production in 2019 is reported as 87,236,221 tons from 4,717,384 hectares of orchard lands [1]. China, the United States, and Turkey are the greatest apple producer countries in the world [2]. In the production of apples, which have such a wide usage area, some productivity-enhancing processes will offer significant gains to the producer. For this purpose, some inspections should be made about the condition of the fruits before harvest in apple orchards. Accurate estimation of yield and detection of diseased apples, especially before commercial agreements are made, is an important issue in terms of preventing major damage. Traditionally, farmers are done this by observing over long periods and roughly inspecting each tree. This process takes a lot of time and causes losses up to 20–30%. Micro-Aerial Vehicles (MAVs) can assist farmers with such inspections. Today, MAVs are used as a solution to many inspection problems (the surface of a dam, bridge pillar, building facade, etc.). With MAVs equipped with camera systems and supported by artificial intelligence, it may be possible to explore the garden corridors and make a more efficient inspection. However, since apple orchards are usually covered with a thin sheet, satellite signals are not efficient and image processing techniques are insufficient for processing dynamic images due to light refraction. In addition, since there are narrow corridors between trees, trajectory control in apple orchards is a very important problem for MAVs. Therefore, the positioning of MAVs for inspection work in apple orchards, image processing and determination of the number of products in the acquired image, and trajectory control are potential problems that need to be resolved.

In this context, studies of image processing and determining the number of apples in apple orchards are examined to calculate apple yield. Image processing techniques are frequently used for object detection [3,4,5,6,7,8,9]. However, image processing techniques can produce low-accuracy and variable results and have problems predicting future unknown data. Especially in the images obtained, the numerical value difference of the pixels caused by the smallest light difference is one of the biggest problems of the image processing method. To overcome these problems in image processing techniques, especially in recent years, object detection and counting based on artificial intelligence have been widely carried out. It is seen that the studies carried out to calculate the yield in apple orchards with artificial intelligence focus on estimating the number of apples [10], determining the number of apples on video [11], and determining the locations of apples with binocular localization [12]. However, in these studies, it has been determined that apples are counted by guessing or more than once on the same video. In addition, deep learning-based approaches, single and two-stage detection techniques are used for object detection. Especially in fruit detection, there are studies in which YOLO [13,14,15] and CNN [16] architectures are frequently used. A detailed comparison of these studies is given in Table 1. However, the common features of these studies are the use of photographic images on the computer for object detection. In matters such as cumulative counting of the detected apples and determining the yield, there is no work on an autonomous system with a custom database.

On the other hand, there is no study on the positioning of the robot, the planning of the robot’s trajectory, and the determination of the number of apples by flying in the determined trajectory, especially in apple orchards.

In this article, the MAV system was developed to increase profitability in apple orchards; The performance of counting apples in the specific trajectory with a deep learning technique is described. This system can perform a process that traditionally takes a very long time and has a high error rate, faster, and with higher accuracy. In this way, producers will be able to calculate the yield more accurately before commercial agreements. In addition, companies that take out insurance for agricultural lands will be able to make more accurate price calculations for their policies by taking advantage of this system. For the said system, the concept of FRS is preferred instead of artificial intelligence-supported MAVs, and it will be mentioned in this way in the later section. An apple orchard is determined in the Yeşilhisar district of Kayseri for the experimental studies of the robotic system. The developed robotic system follows the desired trajectory between the corridors of apple trees, and then the apples in the images obtained with the robot’s camera are detected and counted cumulatively after a specified line. As far as is known, such a study is not done so far, especially in this field. With the results obtained from this experimental study, it is determined that a 10% accuracy rate increase and at least 6 times time savings can be achieved in yield estimation in apple orchards.

## 2. Materials and Methods

Experimental studies are carried out in an apple orchard of 350 decares located in Yeşilhisar district of Kayseri province in Türkiye (38.293621, 35.122010). In this orchard, which has 90,000 apple trees in total, 5% pink lady, 5% granny smith, 10% golden, and 80% red (red chief, scarlet spur oregon spur) apples are grown and it is expected to produce 1700 tons of apples at full yield. In this experimental study, the focus was on red apples and a 50-m section of a corridor with 40 trees is used in the garden.

In this section, the developed system is explained in detail. Firstly, information about the hardware and software specifications used is given. Then, the collection and processing of the data and the training of this data in the CNN network are emphasized. Finally, autonomous trajectory tracking, object detection, and object counting methods of the Flying Robot System (FRS) are explained.

The general methodology of this article is given in Figure 1. As seen in Figure 1, the developed structure consists of 5 basic parts. The first of these parts is the determination of hardware and software specifications. In this part, the equipment to be used for the FRS and the systems that will connect the hardware equipment with the software are introduced. In the second part, the data collection and handling process is explained so that the data can be trained on the CNN network. Then, the training of the processed data with the CNN algorithm is given in the third part. In the fourth part, the planning of the trajectory of the FRS is explained. In the last part, the system developed to detect and count the apple data obtained by the FRS is given. All these parts are explained one by one in the next sections.

### 2.1. Hardware and Software Specifications

In this section, the main frame of the micro aerial vehicle and the systems which are employed are explained in detail, main as shown in Figure 2. The experimental system consists of 2 main parts the ground station and the FRS. FRS; consists of drive systems, avionics, and an external controller with visual aid. As an external controller, the Odroid XU4 mini-computer is used [17]. This mini-computer specifically performs data collection. A laptop is also used as a ground station. FRS communicates with the ground station via radio frequency (RF) signals using the micro air vehicle link (MAVLink) protocol [18].

The aim of the FRS developed within the scope of this study is to detect and count the apples while following the trajectory determined in the apple orchard as shown in Figure 3. In this direction, the robotic system should incrementally count the apples it detects between the 1st and 2nd positions in Figure 3.

Table 2 presents the relevant physical characteristics of the robotic system and the camera mounted on it. The robotic system is driven by 4 KDE-Direct brushless DC motors [19] that can draw a max 24 amper current. For the control of these engines, there are 4 piece 40 amper Electronic Speed Control (ESC) controllers mounted under the arms of the aircraft. A 2500 mah 4 S lithium polymer battery is used to energize this entire flight avionic system. The propulsion system of the aircraft is configured in this way. Pixhawk Cube 2 [20] is preferred because it shows a more stable operation for the control of the flight system. Dronekit library is used to manipulate the robotic system by using the codes developed with Python language over the ground station.

The robotic system follows the autonomously determined trajectory in a corridor determined for experimental work in the apple orchard. Images obtained with the camera for apple detection are transmitted to the ground station equipped with Intel i7-2600 [21], and GTX 1050 Ti [22]. Convolutional neural network estimation for target detection is performed on the ground station. Thus, the hardware weight on the robotic system has been reduced and it has been made more efficient in terms of flight time and energy consumption. In addition, higher-capacity operations can be performed more practically at the ground station. Technical specifications of the ground station are given in Table 3.

### 2.2. Data Collection and Processing

To train the neural network model in the apple orchard determined in Yeşilhisar district of Kayseri province, image data are collected every 2 weeks between September and November of 2021, using the camera integrated on the drone. The GoPro Hero 9 [23] action camera, whose specifications are given in Table 2, is used to obtain images. The resulting images have a resolution of 5120 × 2160 pixels and 20 MP. All images have been reduced to 1728 × 1296 because it would take too long to process with this size. Since FRS will perform object detection at an altitude of 2 m from the ground, the data are obtained on a straight track determined as 50 m at this height with a tolerance of ±10 cm. After the obtained data is stored, it should be labeled for object detection. LabelImg python script, whose interface is given in Figure 4, is used for the annotation process.

The LabelImg script, which is used for labeling data, is a very easy-to-use labeling program. The output of the photos given to the LabelImg script is in .xml format. By using the LabelImg script program, all data is labeled before the object detection process and data in .xml format are created. It is then converted to .record format for use in training the neural network model. To increase the success of the object detection process, the image quality and number must be high. For this reason, 100 photographs are used for object detection. There are, on average, 40 apples in each of these used photographs. Thus, 4000 labeled apple photos are created in the data set. As a result, a custom data set is obtained. This approach will ensure the neural network model makes predictions with higher accuracy.

### 2.3. Convolution Neural Networks

Convolution Neural Networks (CNN), which is one of the Deep Neural Network models, is the most used model in image detection and allows to obtain successful outputs. In Figure 5, the CNN structure is given with its basic features. CNN structures contain four layers: the convolutional layer, the pooling layer, the flattening layer, and the fully-connected layer. In the convolutional layer, various filters are implemented to the image to extract the main features of the image. In the next layer, important features are developed, and the image size is reduced, the flattening layer prepares data as inputs of the neural networks and finally classification is performed in the fully-connected layer. Since the number of parameters that need to be trained in CNN is low, the training is quite fast.

In this study, 20% of the 4000 data in the created data set was reserved for testing and 80% for training. The training data given to the CNN model is trained with 3000 iterations. Figure 6 shows the sample data used to train the CNN model.

### 2.4. Trajectory Planning

In the trajectory planning step, the trajectory that the FRS will follow while performing the task in the apple orchard and the creation of the flight program is carried out. For this, the widely used Dronekit library is used. A laptop is used as a ground station. With the script created in Python, the connection of the FRS with the ground station, the transmission of the commands to be executed, and the monitoring of the data received from the robotic system on the screen are provided as shown in Figure 7. In the robotic system, the position of the robot can be determined from the satellite and tracked from the ground station thanks to the GPS module on the robot.

In the study, firstly, a connection is established between the robotic system and the laptop computer. With the GPS module and the script created, the robot is advanced 50 m along the determined corridor in the apple orchard. Then, the video taken by the robot while the robot is moving is transferred to the computer through the script used. Thus, the image of the apples obtained along the determined trajectory is transferred to the computer environment, and it is ensured that the processes of detecting the apples and determining their number are carried out.

### 2.5. Object Detection and Counting

General object detection is the detection of the object’s location by deep learning methods using a rectangular bounding box. The most widely used deep learning method in object detection is the CNN model. The commonly used object detection scheme is given in Figure 8.

In recent years, CNN is made rapid progress. In general, CNNs are classified as region-based (R-CNN) or regression/classification-based methods (SSD, YOLO, Mobilenet-SSD, etc.) [24]. In this classification, each method has its advantages and disadvantages. Considering both the speed factor and accuracy in object detection, Mobilenet-SSD [25] and Faster R-CNN [26,27,28,29] are two highly preferred models.

MobileNet is a small and efficient CNN network developed by Google. The basic idea of MobileNet is to replace standard convolution with deeply separable convolution, which is a combination of depth direction convolution and point convolution. Thus, the number of parameters is greatly reduced compared to standard convolutional meshes with the same depth meshes. And SSD directly uses layers in the feature pyramid to quickly recognize objects at multiple scales. Thus, a more efficient network structure is obtained by combining the two CNN network features for object detection.

Faster R-CNN is a two-phase zone-based detector and is one of the leading algorithms used in object recognition. There are two stages in Faster R-CNN. The first of these stages is to select the relevant regions from the image. In the second stage, correction and classification of the coordinates of the object are carried out.

In this study, the Tensorflow object detection API package is used to detect objects based on deep learning. As the neural network model, Mobilenet-SSD and Faster R-CNN models, which are pre-trained models in the COCO dataset [30], are used. These models are customized for work in the apple orchard with the FRS. Parameters and values are arranged in accordance with the operation. And training is carried out with the data set produced for this study.

When the apple is detected in an image, CNN marks the image with a border box on the detected apple. This bounding box contains information about how many percent of the apple is similar to the data in the database and how many apples are in that image. In Figure 9, the bounding box given depending on the presence of the apple in an image is shown.

## 3. Results

In this section, the experimental results obtained from FRS and the performance of the developed artificial intelligence model are presented. Figure 10 below shows the result of apple detection with bounding boxes and accuracy levels using 4 different CNN models. Figure 10a shows the result of detecting apples on the video image obtained from FRS with the SSD-Mobilenet neural network model trained with the custom dataset produced. On the other hand, as shown in Figure 10b, the performance ratio is very high in the experiment performed with the Faster R-CNN model working with double scanning for detection. Figure 10c,d show apple detection study results obtained with SSD-Mobilenet and Faster R-CNN models trained with the standard COCO dataset [30]. The results are extremely low, as expected.

Table 4 shows the comparative performance results of the neural network models developed using two different CNN model architectures in this study. Parameters used in training SSD-Mobilenet and Faster R-CNN models developed within the scope of this study; Training steps, batch size, and learning rate were set to 3000 and 4, with a variable in the range of 0.015–0.04, respectively.

Performance metrics including the F1 score are used to quantitatively more accurately evaluate apple detection performances. Results are divided into four types, true positive (TP), true negative (TN), false positive (FP), and false negative (FN), based on the relationship between the true object class and the predicted object class. Then, precision (P) and recall (R) are defined in Equations (1) and (2):P = TP/(TP + FP)(1)
R = TP/(TP + FN)(2)

Then, the F1 score is defined based on the P and R values as in Equation (3):F1 = 2.P.R/(P + R)(3)

Mean average precision (mAP) is used for detection accuracy depending upon the precision and recall and averaged for an overall score. For the quantitative measure of the localization, Intersection over Union (IoU) [31], an application of the Jacared index, is computed as:(4)$$IoU\;(b^{pred},b^{g})=\frac{Area(b^{pred}\cap b^{g})}{Area\;{(b}^{pred}\cup b^{g})}$$
where $b^{g}$ is the bounding box of labeled ground-truth and $b^{pred}$ is the predicted bounding box from an object detection model. The *IoU* threshold is used as a boolean operator to eliminate the false-positive bounding boxes with smaller *IoU* scores. This *IoU* checks the intended sensitivity for the localization to be negative or positive (i.e., *IoU* ≥ threshold). Different threshold values are used such as 0.25, 0.5 and 0.75 for model evaluation [32].

Accuracy means the proximity of a measured value to a standard value as in Equation (5):Accuracy = (TP + TN)/(TP + TN + FP + FN)(5)

A total of 140 apples from 8 trees were included in the artificial intelligence-assisted apple detection and counting application performed at the ground station over the images obtained from the FRS. The number of apples detected by the proposed method; is 129 for the Faster R-CNN model and 65 for the SSD-Mobilenet model. In addition, the number of false detections and F1 scores is shown in Table 5 comparatively.

Loss function *l* is used to learn to find out the optimal features required to enhance the model and can be represented as;
(6)$$l(x,a)=\sum_{i=1}^{N}l\left(y_{i},x_{i},b^{pred};a\right)+\frac{\lambda }{2}{\left||a|\right|}_{2}^{2}$$
where a is the deep learning parameter obtained from the training process and λ is the regularization parameter.

Figure 11a shows the change in learning rates of the trained SSD-Mobilenet and F-RCNN models. The low learning rate in the SSD-Mobilenet model can be concluded that the estimator is stuck at the undesirable local minimum. Figure 11b presents the loss values of both models comparatively. It can be said that it is related to the fact that the F-RCNN model gives more successful results in estimating the loss value in training.

Figure 12 shows the values describing how many steps per second (as an average value for every 100 steps) are completed for both models (SSD-Mobilenet, Faster R-CNN) while the neural network models are trained on the CPU at the ground station. It can be easily seen from the results that the training of the F-RCNN model takes longer due to the double detection process.

## 4. Discussion

Since this study aims to detect and count apples from the images obtained from the camera on a micro aerial vehicle, it is necessary to first find out which approach would yield more accurate results on images that are exposed to moving and disruptive effects. First of all, it is proved why the widely used ready-trained models aren’t preferred. In particular, the importance of training the neural network model well with the specific dataset in the studies on moving images in the real field environment is understood.

The reason why SSD-Mobilenet and Faster R-CNN models are preferred to examine their performance is; that studies have shown that the SSD-Mobilenet approach offers a balanced performance in terms of speed and accuracy for mobile systems, while the Faster R-CNN approach performs slower but more detailed scanning. The fast and energy-efficient approach may be promising in future studies in terms of performing this operation on FRS. On the other hand, the use of a more accurate approach by transmitting data to the ground station, although slow, can also provide reliable results. In addition, the slower operation of the F-RCNN model creates a disadvantage for FRS. This situation was tried to be overcome by controlling the speed, but this aspect of the model can be improved for further studies.

## 5. Conclusions and Future Directions

In this experimental study, commonly used pre-trained two models with Faster R-CNN and SSD-Mobilenet architectures and optimally adjusted versions of hyperparameters trained with custom dataset using these two architectures compared to performance in apple detection and counting application using FRS. According to the results obtained from the experiments with CNN models that offer different approaches; It has been concluded that for an application where the accuracy of the results is important, such as yield estimation, it is more appropriate to choose the Faster R-CNN approach that performs high-precision processing. SSD-Mobilenet and Faster R-CNN models trained with the custom dataset showed higher performance than other models. The loss value of the specially trained Faster R-CNN model has decreased below 0.1 and it can make predictions with an accuracy rate of 93% (with 3% tolerance). It is expected that this ratio will decrease further with the correction functions to be performed.

The Faster R-CNN model, which is developed by training with the custom dataset, has reached a high accuracy rate, although it needs higher performance for its operation. As a solution to the performance need; A solution was found by deciding to transfer the collected data to the ground station and process it there. This situation does not pose a significant problem in terms of the functioning of the study. As a result, promising results have been obtained in the field of control studies with automatic FRS in apple orchards. In future studies, improvements can be made to count the detected apples cumulatively without mixing with each other. Counting objects from the image by separating the unique frames in the video can improve accuracy. In addition, it is recommended to perform studies on location verification with the SLAM technique and energy efficiency to count the entire garden. In this way, it will be possible to prevent problems caused by distortions in satellite signals and position errors.

## Figures and Tables

**Figure 1 sensors-23-06171-f001:**
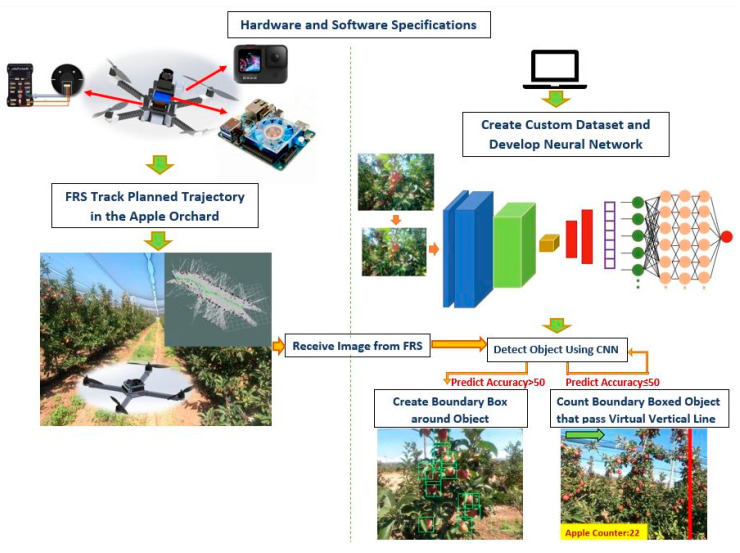
A general methodology of the proposed system structure.

**Figure 2 sensors-23-06171-f002:**
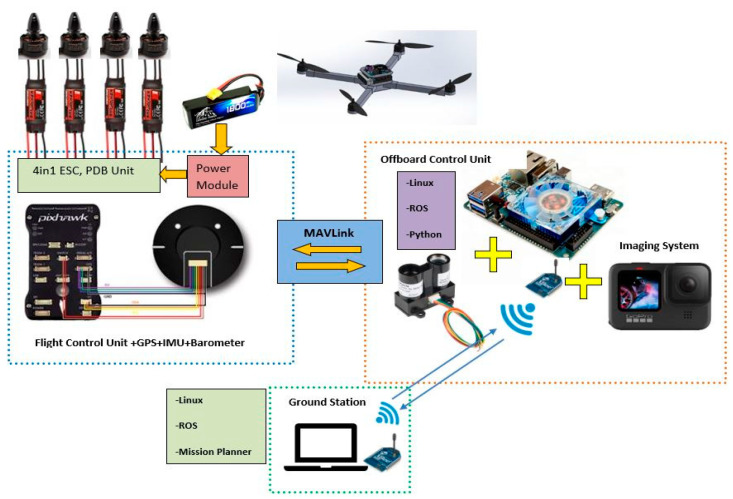
Flying Robotic System (FRS) basic architecture.

**Figure 3 sensors-23-06171-f003:**
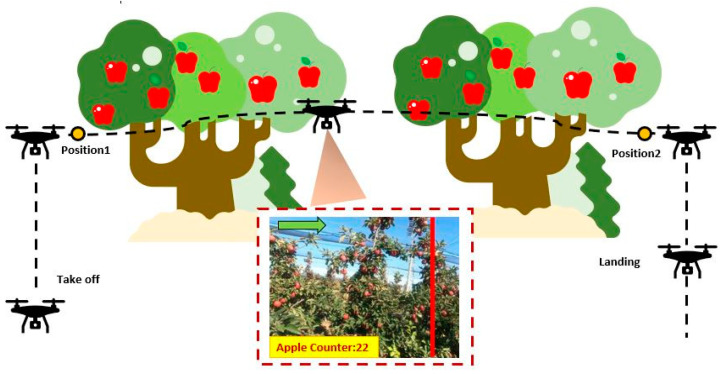
Trajectory tracking planning of FRS in the apple orchard.

**Figure 4 sensors-23-06171-f004:**
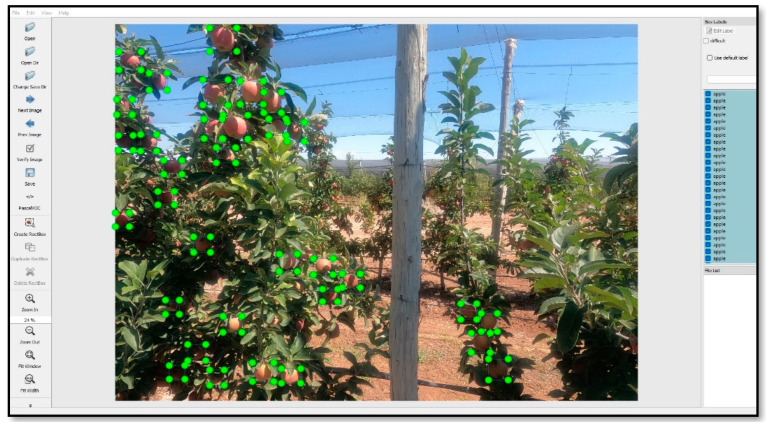
The interface of the LabelImg python script is used for annotation.

**Figure 5 sensors-23-06171-f005:**
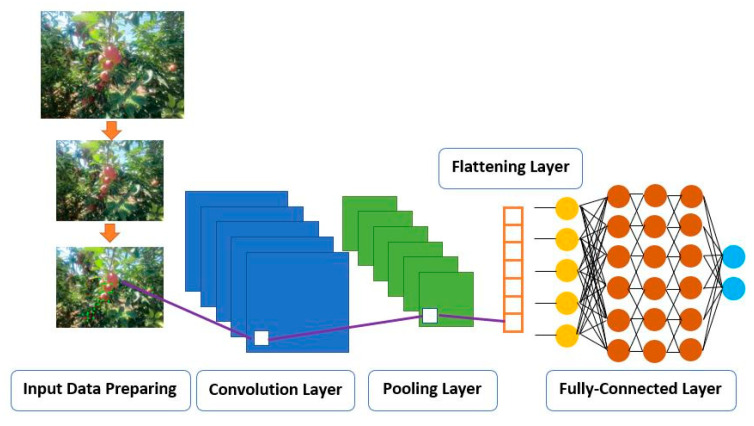
CNN general structure.

**Figure 6 sensors-23-06171-f006:**
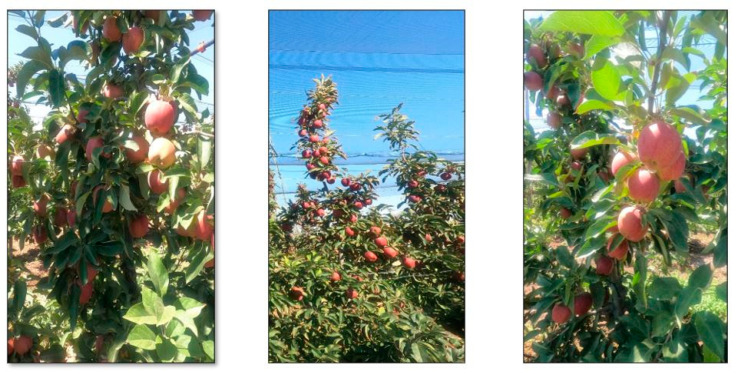
The sample image data is used to train the CNN model.

**Figure 7 sensors-23-06171-f007:**
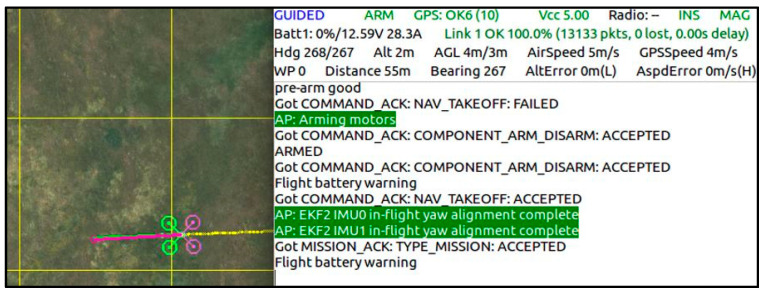
Flight planning program interface.

**Figure 8 sensors-23-06171-f008:**
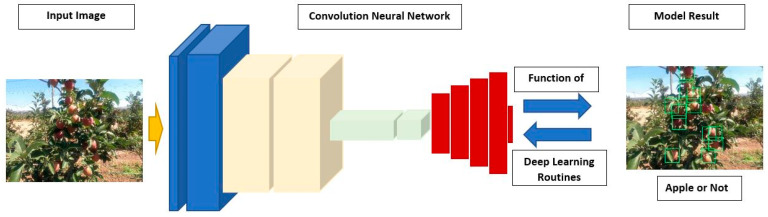
Convolution Neural Network Object detection scheme.

**Figure 9 sensors-23-06171-f009:**
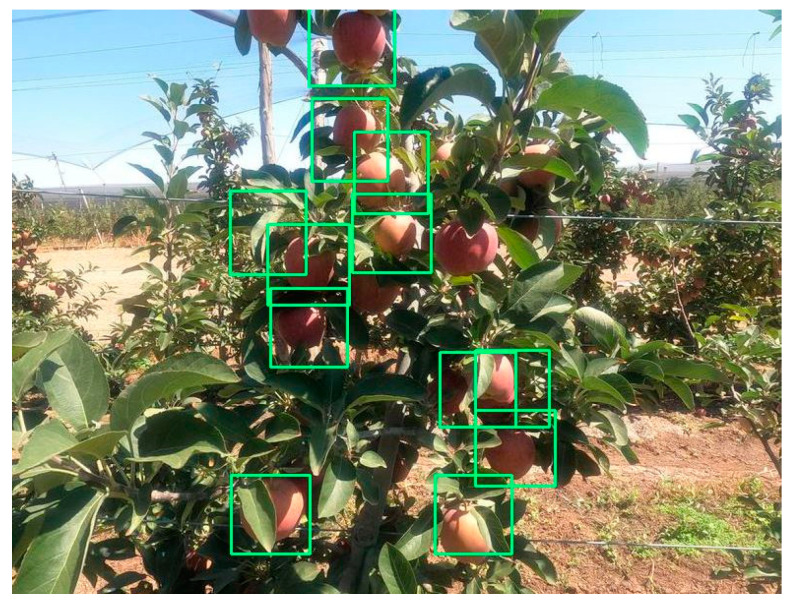
The image of apples bounded by the box as a result of the object detection process performed on the test image data.

**Figure 10 sensors-23-06171-f010:**
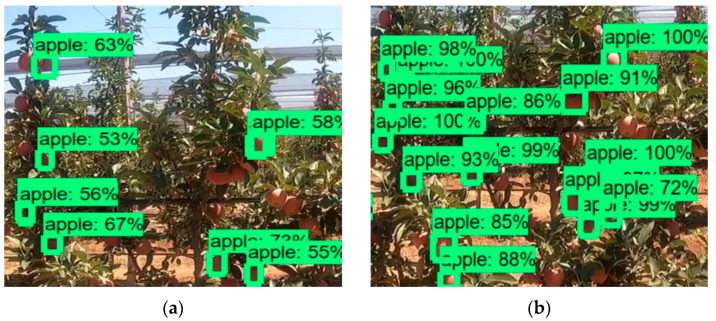
The result of detecting apples on the video image with (**a**) the SSD-Mobilenet neural network model trained with the custom dataset (**b**) trained Faster R-CNN (**c**) standard SSD-Mobilenet (**d**) standard Faster R-CNN.

**Figure 11 sensors-23-06171-f011:**
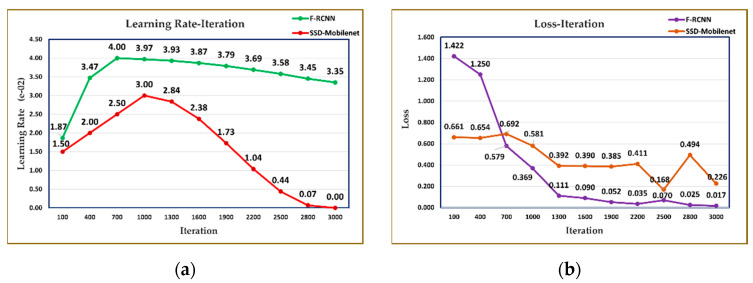
For SSD-Mobilenet and Faster R-CNN models; (**a**) Learning rate- Iteration, (**b**) Loss-Iteration graph.

**Figure 12 sensors-23-06171-f012:**
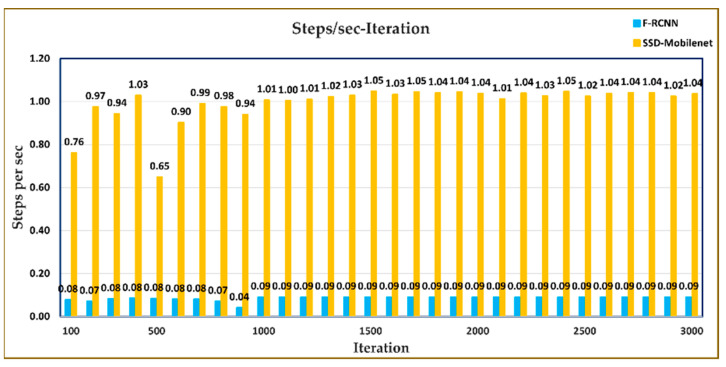
Steps per second-Iteration graph for SSD-Mobilenet and Faster R-CNN models.

**Table 1 sensors-23-06171-t001:** The comparison of related studies.

Reference	Subject	Model Architecture	Video/Image	Robotic Application
[13]	Pear and Apple Detection	YOLO v2	Video	No
[14]	Tree Inspection	CNN	Image	No
[15]	Fruits Counting	YOLO v3	Video	No
[16]	Weeding	YOLO v3	Image	Yes
This study	Apple Counting	F-RCNN	Video	Yes

**Table 2 sensors-23-06171-t002:** Physical characteristics and specifications.

System	Technical Details	Description
Robotic System	Configuration type	X-Config
	Dimension	45 × 45 × 25
	Payload	Total 3 kg
	Durance	15 min
	Brushless Motor	KDE Direct
	Electronic Controller	HobbyWing 40 A 620 Hz
	Propeller	11 × 4.7 CCW/CW Nylon
Camera	Resolution	20 Megapixel
	Pixels	5120 × 2160
	Communication	Mini USB

**Table 3 sensors-23-06171-t003:** Technical specifications of the ground station.

	Title 2
GPU	NVidia
CPU	Intel i7
Memory	16 GB DDR4
Operation System	Windows 10 Pro

**Table 4 sensors-23-06171-t004:** Precision, Recall, and F1 score values in apple detection for trained two methods.

Method	Recall	Precision	F1 Score	mAP@0.5
SSD Mobilenet (proposed)	0.72	0.83	0.77	71%
F-RCNN (proposed)	0.87	0.96	0.91	93%

**Table 5 sensors-23-06171-t005:** Results of the apple counting process and average accuracy.

Method	Detected Number	Wrong Detection	Accuracy
SSD Mobilenet (trained)	65	2	46%
SSD Mobilenet (COCO)	32	8	13%
F-RCNN (trained)	129	1	93%
F-RCNN (COCO)	10	12	6%

## Data Availability

The data presented in this study are available on request from the corresponding author.

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
