# Peer review of "Deep Learning Based Apples Counting for Yield Forecast Using Proposed Flying Robotic System"

_sensors, 2023, doi:10.3390/s23136171_

Round 1

Reviewer 1 Report

This work employed two different neural network model estimators to detect apples using the Single-Shot Multibox Detection (SSD) Mobilenet and Faster Region-CNN (Faster R-CNN) model architectures. This is done with the custom dataset generated from the red apple species (Red Chief, Scarlet Spur, Oregon Spur). The authors compared experimentally in terms of performance the SSD-Mobilenet and Faster R-CNN architecture models trained with COCO datasets referenced in many studies and SSD-Mobilenet and Faster R-CNN models trained with the custom dataset prepared for apples. The paper's contribution to existing knowledge in this research field is well justified. The authors mentioned some recent techniques, but the paper has not addressed the motivation for developing another method. The following points can improve the manuscript.

  1. Add some numerical results to the abstract.
  2. Enhance the introduction to show the motivation of this work.
  3. A comparative study can be added to the related work section in table form to show the recent efforts.
  4. Change the subtitle " 2.3. CNN” to "2.3. Convolution Neural Networks”
  5. Figure 4 is part of Figure 2. Please remove it from the manuscript.
  6. Figure 6 is part of Figure 1; update one of them.
  7. Double-check all equations to be true. (compulsory)
  8. Performance evaluation metrics are not enough. Add some other metrics and explain them all mathematically.
  9. The proposed method should be compared with recent techniques.
  10. There should be some discussions on the limitations of the presented methods in a separate section. 
  11. Change the “Summary” section title to “conclusion and future directions” and add more future directions to the research.
  12. Enhance the English of the work. There are too many problems with paper typesetting.
  13. The paper is not suitable for acceptance in its current form. The article needs rewriting to address the comments mentioned above. 

Reviewer 2 Report

I believe that your work is in a mature level to be published. 

The authors could provide some more references within the introduction section. Also, due to many abbreviations, authos could add a table with all of them.

Author Response

Please see the the attachment

Reviewer 3 Report

This paper is well written. The novelty and contributions are sufficient. I suggest some minor comments to improve this paper before acceptance.

1. Please discuss the motivation of this study in detail.

2. Please add more recent references in the LR section and review CNN critically.

3. Any comparison could be made between the proposed and existing designs?

4. Please elaborate on the practical implications of this study in the conclusion section. 

Round 2

Reviewer 1 Report

The authors have addressed most of my concerns. The paper can be accepted.